# Playgrounds for Abstraction and Reasoning

**Subin Kim**      **Prin Phunyaphibarn**      **Donghyun Ahn**      **Sundong Kim**[*]
School of Computing, KAIST                            AI Graduate School, GIST
21supersoo, prin10517, segaukwa@kaist.ac.kr          sundong@gist.ac.kr

## Abstract

While research on reasoning using large models is in the spotlight, a symbolic method of making a compact model capable of reasoning is also attracting public attention. We introduce the Mini-ARC dataset, a 5x5 compact version of the Abstraction and Reasoning Corpus (ARC) to measure the abductive reasoning capability. The dataset is small but creative, which maintains the difficulty of the original dataset but improves usability for model training. Along with Mini-ARC, we introduce the O2ARC interface, which includes richer features for humans to solve the ARC tasks. By solving Mini-ARC with O2ARC, we collect human trajectories called Mini-ARC traces, which will be helpful in developing an AI with reasoning capability. Mini-ARC is available at `https://bit.ly/Mini-ARC`.

## 1   Mini-ARC: 150 Tasks for Measuring Intelligence

The first part of the paper introduces the Mini-ARC dataset, a set of visual reasoning tasks [1] that requires a complete understanding of human priors to solve each task correctly. To make the dataset, we set up certain principles that satisfy this purpose introduced in §1.1.

### 1.1   Principles

To solve the original ARC problem proposed by Francois Chollet [2], we need to develop a solver with the ability to handle input/output of various dimensions (1x1 to 30x30), in addition to the reasoning ability. This seems to be an unnecessary burden in the early stage of research. We curate a Mini-ARC data set that fixes input/output to 5x5 in order to reduce the modeling budget and concentrate on developing a solver close to that of human cognition. The reason for choosing the size 5x5 is as follows.

- When width and height are odd, the grid space has its central point.
- Square shape will encourage using basic primitives such as rotation to solve the task.
- 1x1 and 3x3 are too small, so rotation and symmetry were difficult to distinguished.

We invited 25 colleagues with a basic understanding of ARC to generate novel 5x5 tasks, including at least four demonstration pairs (input-output pairs). We instructed participants to build a task with a clear and unique solution to fit the intent of the ARC problem in this abbreviated size. During two day event, they spent two hours solving ARC problems and four hours creating new problems. As a result, we collected about 150 Mini-ARC tasks. Figure 1 illustrates the data collection interface.

As an example of this, various levels of problems can be created within 5x5 with different levels of complexity, such as an easy task consisting only of primary functions such as rotation and flip as the examples in the original research [2], or a difficult task where average description recorded in LARC [3] is lengthy.

---

[*]Corresponding author

36th Conference on Neural Information Processing Systems (NeurIPS 2022).

Following the principles of ARC, participants are not encouraged to use prior external knowledge such as drawing alphabet in the grid space. We also maintain the color option for each pixel same as the original ARC data set. The instructions were structured to respect the creativity of individual generators as much as possible.

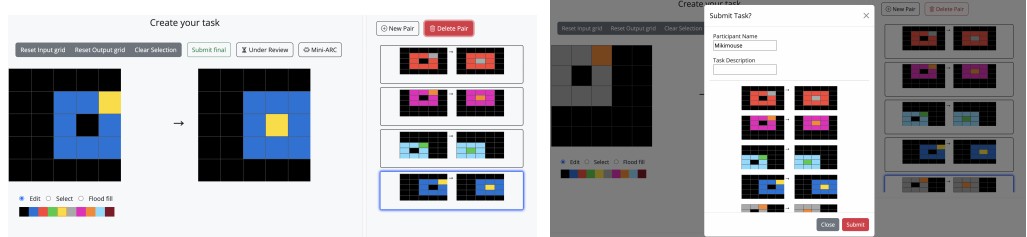

(a) Task generation screen        (b) Task submission screen

Figure 1: Interface for curating the Mini-ARC dataset. We allow participants to check other submissions while designing their tasks. Submitted tasks are queued in the under-review list. Administrators verified each task and notified participants about the approval. Resubmitting after editing the already made tasks is allowed. Finally, 150 approved tasks consist of the Mini-ARC dataset.

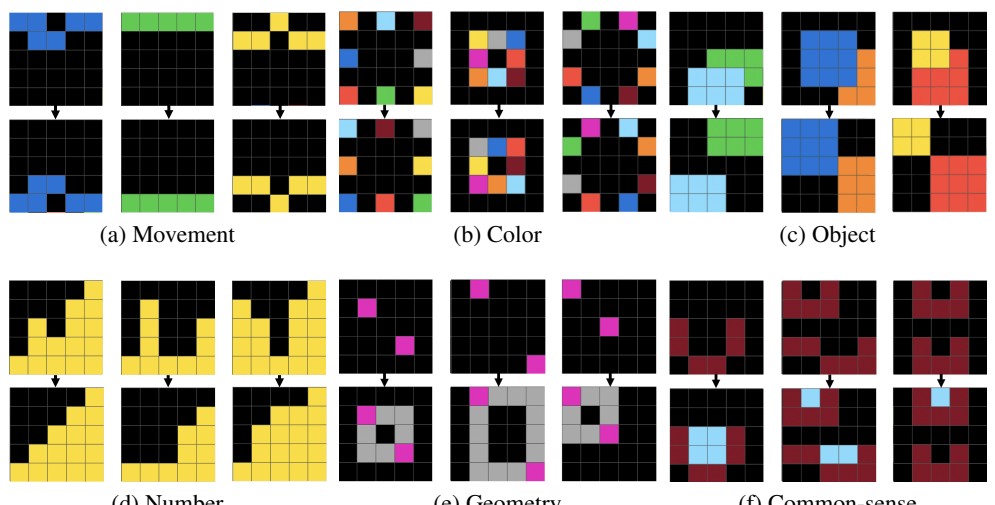

(a) Movement      (b) Color      (c) Object

(d) Number      (e) Geometry      (f) Common-sense

Figure 2: Representative Mini-ARC examples for each category

## 1.2 Categorization

Chollet have proposed the ARC problems based on the four broad categories (Objective, Geometry, Agentness, Number) of innate assumptions that form the foundations of human cognition identified by core knowledge [2]. We divided the Mini-ARC tasks into six categories based on these priors and the key ideas for each task: movement, color, object, number, geometry, and common sense.

**Movement** tasks follow the rules based on dynamic modifications such as flip, rotation, and sliding sideways. The example shown in Figure 2(a) can be solved by flipping the 5x5 grid horizontally.

**Color** tasks are highly dependent of the color aspect of each pixel, such as swapping colors. For example, the color of the 1x1 pixel rotates counter-clockwise in Figure 2(b).

**Object** The movement of the object or agent plays a vital role in solving the problem. An object refers to an area consisting of two or more adjacent pixels that can intuitively distinguish the object from the background.

**Number** tasks could be solved using the concept of numbers, such as counting the number of pixels of the same color and sorting the pixels accordingly or modifying the pixels so that the number of pixels could vary.

**Geometry** tasks require the concept of geometric structures. The problems involved mainly unifying the color of grids of the same row or column or swapping the same colored areas into other colors.

**Common-sense** tasks, like maze-pathfinder, or Tetris, require high-level induction, although it may be intuitively evident for humans. Most of these problems do not fall into previous categories but are based instead on common sense. For example, The input-output pairs in Figure 2(f) demonstrate rain drops, where brown-colored grid serves as a container.

## 1.3   Quality Evaluation

Contributions from 25 participants allowed us to construct Mini-ARC tasks as sufficiently original problems—Figure 3(a), average 2.59. In order to rate the novelty and difficulty of the task, participants analyze the created data set and assign a score between 1 and 5. Two grading scales are used, one for human solvers and the other for building AI models. A total of 208 responses were collected and each Mini-ARC task received at least one evaluation.

**Difficulty** Most of the respondents evaluated that they could solve a given problem intuitively—Figure 3(b), average 2.02. Participants felt that developing a program to solve each problem was slightly more difficult than solving it directly—Figure 3(c), average 2.12. However, participants thought of developing a task-specific program when evaluating each problem. The difficulty of developing a general solver is expected to be evaluated much higher.

**Comparison between human and program** Figure 3(c) shows the result of subtracting the score of difficulty for humans from the score of difficulty for computers for each evaluated problem set. If the difficulty for computer is greater than the difficulty for humans, it reflects the induction gap between the program and the human. 64 out of the 208 responses (31%) claimed that implementing the program was more difficult than solving the task manually. 104 respondents (50%) claimed that implementing the program is as difficult as solving the task.

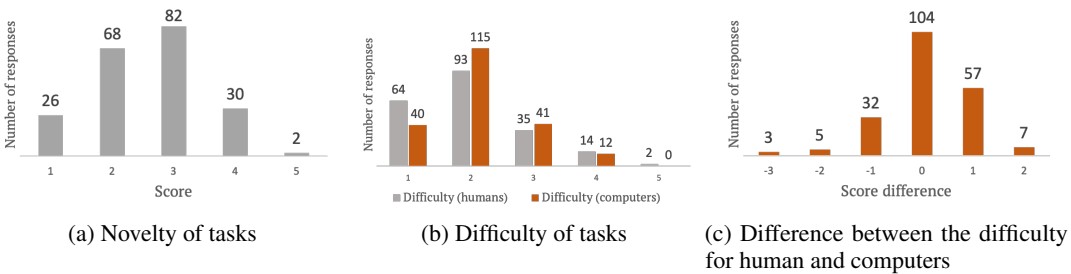

| (a) Novelty of tasks | (b) Difficulty of tasks | (c) Difference between the difficulty for human and computers |
|---|---|---|

Figure 3: Survey results of Mini-ARC

## 1.4   Challenges and Suggestions

**Challenges** When curating the Mini-ARC dataset, we limited the size of the input and output to 5x5, which might restrict task designer's creativity. Therefore, it is challenging to measure general intelligence by only using Mini-ARC. This can be seen as an opportunity cost inevitably borne when space is constrained. After finding a way to solve the current Mini-ARC effectively, it will be possible to expand and switch to a form that does not restrict input and output grid size.

**Suggestions** There are many ways to utilize and verify Mini-ARC dataset. Since there are at least four input-output pairs in each task, several combinations of demonstration and test pairs can be made in the training process. We can measure the soundness of the problem by comparing the patterns from these pairs. In addition, we can compare the characteristics of each category while developing its solver.

## 2 O2ARC: Tools for Collecting Expert Demonstrations

We re-design the browser-based interface for humans to solve the ARC tasks with their hands. We name it Object-Oriented ARC (O2ARC). The tool has six additional functions to ease the manual problem-solving process, and we will describe the principles and procedures hereafter.

### 2.1 Motivation of Redesigning the Original Tool

The functions included in the tool provided with the original ARC dataset were pretty basic. It contains three symbol controls: edit, select, and flood fill, and three grid controls: copy from input, resize grid, and reset grid to null.[2] All ARC problems can be solved by using these basic functions, but the lack of convenience makes the participant click the button repetitively when solving the problem. Therefore, the human trajectory saved by this tool is long and messy and often does not reflect the intuition of solving the problem at a glance.[3]

### 2.2 New Features: Six Primitives

First, we brainstorm additional functions grounded that many ARC problems are object-oriented, as illustrated in Figure 2. Inspired by editing tools such as Microsoft Powerpoint and Adobe Photoshop, functions such as cut, undo, redo, flip, rotate, and ctrl+select was implemented and added to the O2ARC tool. Through these functions, O2ARC provided a higher level of convenience. The details of these functions were improved through alpha testing.

### 2.3 Layers

The function we concentrated on the most is a layering function. With this function, the user can save each object in one layer and control the object's position by manipulating the layer instead of modifying the final output grid. The user can change the order of the stacked objects easily by changing the order of the layers. This function enables the user to solve some problems concisely.[4] Aside from the convenience of the original ARC, the layer function was not necessarily needed to solve the 5x5 size of Mini-ARC. Therefore, in unraveling the 5x5 size Mini-ARC, we disabled this function and collected Mini-ARC Trace.

### 2.4 Interface Design

Other than that, we tried to change the layout of the interface by showing an input-output pair one by one instead of showing all training pairs together. This is to collect fine-grained user trajectories to answer some questions: How many input-output pairs do users see before making any action? How many seconds did the user look at each pair? Answering these questions helps us measure the task's difficulty or the wit and agility of the participants. Contrary to the intention, some participants complained that this layout takes away the advantage of observing pairs altogether. Therefore, we restored the original design.

## 3 Mini-ARC Trace: Compiled Expert Demonstrations

### 3.1 Principles of Collecting Mini-ARC Trace

Tasks used for gathering Mini-ARC trace were picked uniformly from six categories. 20 participants were gathered to solve ten of Mini-ARC tasks within two hours, where each set contains five tasks, one from each category. Among six categories, common-sense tasks requires a fairly original solution, thus traces collected from these problems are unlikely to be applied to solve other types of problems. Thus, collecting traces of common-sense problems has been postponed, as our short-term goal is to create an AI model that can identify and imitate frequent patterns found from human experts. If the participant submits three consecutive wrong answers, the system determines that the user cannot find the solution to the problem.

---

[2]Description of these functions is on the ARC repository: `https://github.com/fchollet/ARC`.

[3]See complex state-space graphs on `https://arc-visualizations.github.io/`.

[4]See demonstrations on `https://youtu.be/zo1G4E720Co?t=108`.

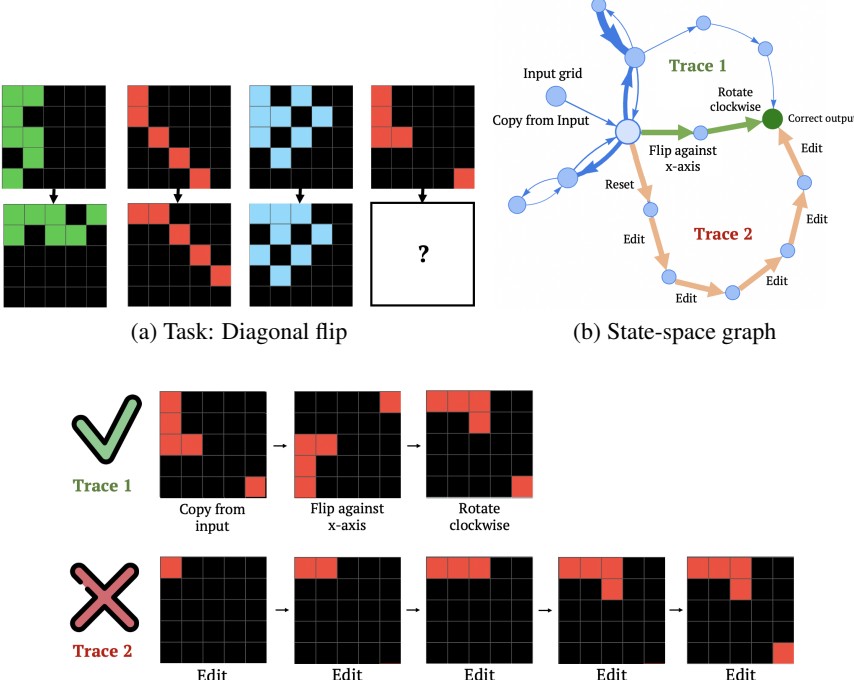

(a) Task: Diagonal flip

(b) State-space graph

(c) Two human traces: One uses only edit button, the other uses flip and rotate.

Figure 4: Different traces are logged for the same diagonal flip problem. Trace 2 using only the edit button is not generalizable to other training pairs, which is not the desired answer. The logic behind Trace 1 is a diagonal flip, which is the correct answer. Selecting meaningful traces and training a model based on them will be the next challenge.

For each Mini-ARC task, we combined all traces to form a state-space graph. Figure 4(b) demonstrates the visualization of multiple traces collected from different participants. A sequence of actions can be identified from each trace. Each node represents the state of the output grid, and the green node represents the correct output grid.

## 3.2 Challenges and Suggestions

**Challenges** Figure 4(a) is a simple diagonal flip task, and we highlight two representative traces in Figure 4(b). As seen in Figure 4(b)–(c), not all traces contain the high-level process of human intuition. For example, in Trace 2, the task is solved through naive action sequences using only edit. The restricted input and output grid size of Mini-ARC may take part in increasing naive actions. However, Trace 1 reflects the intuition of diagonal flips using compact movements, and the process for solving the task is also more effective. The question of how to collect traces that mirror intuition remains.

**Suggestions** The purpose of Mini-ARC trace is to collect reference trajectories for reference for the programs. We can utilize the traces as a replay buffer for training agents through imitation learning. We also plan to analyze the traces based on action sequences to find the new consequent series of actions that can be generally used for solving multiple Mini-ARC.

## 4 Conclusion

We present the Mini-ARC produced as part of challenging abductive reasoning problem [2, 4], and introduce O2ARC, which enables including high-level features of human intuitions when solving ARC tasks. With O2ARC, we collect Mini-ARC traces that can be used to advance the program synthesis [5] and RL agents [6, 7] to move one step toward general intelligence.

## Acknowledgments and Disclosure of Funding

We appreciate everyone who participated in Happy ARC Day, including Sungwon Han, Seung-Eon Lee, Sungwon Park, Wenchao Dong, Sheikh Shafayat, for exchanging ideas with us. This work was supported by the Institute for Basic Science (IBS-R029-C2, IBS-R029-Y4) and the IITP grant funded by the Korea government (No. 2019-0-01842, AI Graduate School Program, GIST).

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
