# OpenReview forum: "Playgrounds for Abstraction and Reasoning"
_NeurIPS.cc/2022/Workshop/nCSI — nCSI WS @ NeurIPS 2022 Poster_

### Official Review · Reviewer_5TA7 · 2022-10-06
**mini-ARC - small Image-in-image-out puzzles with human solve traces**

**Rating:** 2
**Confidence:** 2

**Review:**

## Contributions and Relevance

This paper's core contribution is a novel dataset, mini-ARC, akin to ARC (Abstraction and Reasoning Corpus), consisting of 5x5 input/output image pairs, for tasks such as program induction. Annotations include human solve traces and human difficulty ratings. The size limitation was imposed to lessen the burden on computational resources.
The dataset is easily employable for neuro-symbolic AI; causality was not discussed. There might be an unmentioned connection to causality via the collected traces of human solvers; this could be expanded upon. With minor changes, I think this paper is quite relevant to the workshop.

## Clarity

The text is generally mostly clear and well-written; exceptions include 2 apparently unintended negations.
* 1.2: first paragraph list lacks category "color"
* In subsection on color: s/independent of/dependent on
* In 1.4, the ref{sec:category} broke.
* 3.1: s/cannot not/ can not

## Strong points that could use more detail

As the paper mostly expands upon ARC, its main idea is not novel; the dataset it introduces is though. I can not judge significance of the presented dataset; for that a comparison with all small (e.g. < 7x7) problems in ARC might be helpful, as well as a discussion of runtime/comutational demands on methods currently applied on ARC or comparable problems to justify the size limit. As far as I can tell, miniARC adds significant numbers of samples to ARC, at least in the low-resolution segment, and expands upon ARC by including human solve traces. These strengths could be expanded upon.

The paper could profit from some background on the original ARC. Citation [2](Chollet 2019) should be placed more clearly and prominently. Chollet 2019 talks a lot about theoretical reasons for the individual tasks, are those considerations mirrored in miniarc?

## Weak points / criticism:

2.3/2.4 - these are essentially dropped features, right? Discussion of such can be much more succinct, as they have no bearing on the dataset and just explore the design space.
While those are interesting discussions, they'd need to be longer and supported by additional data to be worthwhile. Better traces and human solver behavior is interesting; the salient question here would be how can we get this regardless of the obstacles outlined in 2.3/2.4?

Possible deanonymization of authors via footnote 3, page 4?

Fig 4: The red X and green check mark symbols suggest that one trace was wrong, when it arrived at the same solution. I'd suggest changing the symbols. E.g. the two traces in (c) could use orange and green background colors to map them to (b). I'd also refrain from implicitly restricting the methods to be applied to the data (... will be the next challenge) and mark that as one future avenue (out of many).

Given more space (after the concerns mentioned above) I would appreciate more discussion on the varied possible methods that are enabled by this dataset.

---

### Official Review · Reviewer_m9M7 · 2022-10-12
**Review of Playgrounds for Abstraction and Reasoning (Mini-ARC)**

**Rating:** 2
**Confidence:** 2

**Review:**

## General
The authors propose the Mini-ARC dataset, which is inspired by the ARC dataset by F. Chollet (2019) but every task has a small, fixed grid-size. They performed a study with humans to create ground-truth “traces” by use of their specific interface.
They used a form of crowd-sourcing to come up with 150 problems from multiple categories. Their goal is to have a compact dataset to enable models learning to reason in human ways.
Quality: The quality is good, they used multiple people to come up with the tasks which should lead to more creativeness and they have the user studies to define a human ground-truth. A baseline for the task would further strengthen the paper.
Clarity: The paper is written understandably, the figures are helping but could use some tweaking to improve readability (e.g. font size).
Originality: Both high and low at the same time, since it’s mainly an extension to ARC-dataset with fixed small grid-size. Still, there are 150 extra tasks, while original ARC is 800 (train+eval). Access to more quality datasets is generally desirable.

## Suggestions
To further strengthen the paper I would suggest to a) add a baseline, b) add statistics describing the dataset (e.g. how many steps are needed to solve puzzles) and c) since the work relies heavily on the ARC dataset, to draw a comparison to it and show how/if they integrate.

## Pros:
+ new dataset (or dataset extension) for reasoning tasks
+ clarity: paper is well written, they do describe well what they did
+ O2ARC interface for easier use for human task takers will be released with the data (or so I understood)
+ human traces from multiple participants
## Cons:
- They excluded the common-sense category when generating human traces, while this to me seems to be one of the most interesting categories. They argue that they might be too hard for the humans, however, in their study the users tended to rate the difficulty as rather low, even for humans (Fig. 3)
- Lack of introduction/motivation/background. This would be nice for the reader to give context. The motivation they deliver seems rather weak, they claim the “action space becomes inevitably large”, while not saying/showing that it is *too* large for current SOTA.
- No baseline provided, could especially be interesting since they have their participants evaluate the difficulty of the tasks for humans and AI. Does the baseline reflect this estimation?
## Typos:
Line 44: Typo assumed, should be “highly DEPENDENT”
Line 86: missing \ref

---

### Meta-Review · Area_Chair_uiBV · 2022-10-19

**Recommendation:** 2
**Confidence:** 2

**Metareview:**

The paper key contribution is a nwe dataset  mini-ARC similar to
to the ARC (Abstraction and Reasoning Corpus).
It consists of 5x5 input/output image pairs that can be used in program induction
and applications.
One of the goals of the authors
is to have a compact dataset that enables model learning
to reason in human ways.
Both reviewers suggest that the paper has mostly positive aspects
and deserve acceptance.
I agree with the reviewers and propose acceptance of this paper.

---

### Decision · Program_Chairs · 2022-10-20

Accept (Poster)